# Plastic and Placenta: Identification of Polyethylene Glycol (PEG) Compounds in the Human Placenta by HPLC-MS/MS System

**DOI:** 10.3390/ijms232112743

**Published:** 2022-10-22

**Authors:** Antonio Ragusa, Veronica Lelli, Giuseppina Fanelli, Alessandro Svelato, Sara D’Avino, Federica Gevi, Criselda Santacroce, Piera Catalano, Mauro Ciro Antonio Rongioletti, Caterina De Luca, Alessandra Gulotta, Sara Rinalducci, Anna Maria Timperio

**Affiliations:** 1Department of Obstetrics and Gynecology, Università Campus Bio Medico di Roma, Via Álvaro del Portillo, 21, 00128 Rome, Italy; 2Department of Ecological and Biological Sciences, Università degli Studi della Tuscia, 01100 Viterbo, Italy; 3Department of Pathological Anatomy, San Giovanni Calibita Fatebenefratelli Hospital, Isola Tiberina, Via di Ponte Quattro Capi, 39, 00186 Roma, Italy; 4Department of Obstetrics and Gynecology, Università Degli Studi di Sassari, 07100 Sassari, Italy

**Keywords:** plastics particles, placenta, polyethylene glycol, UHPLC–mass spectrometry

## Abstract

The placenta is a crucial interface between the fetus and the maternal environment. It allows for nutrient absorption, thermal regulation, waste elimination, and gas exchange through the mother’s blood supply. Furthermore, the placenta determines important adjustments and epigenetic modifications that can change the phenotypic expression of the individual even long after birth. Polyethylene glycol (PEG) is a polyether compound derived from petroleum with many applications, from medicine to industrial manufacturing. In this study, for the first time, an integration of ultra-high-performance liquid chromatography (UHPLC) coupled with mass spectrometry (MS) was used to detect suites of PEG compounds in human placenta samples, collected from 12 placentas, originating from physiological pregnancy. In 10 placentas, we identified fragments of PEG in both chorioamniotic membranes and placental cotyledons, for a total of 36 samples.

## 1. Introduction

Plastic, microplastic (MP), and nanoplastic pollution has reached every part of the planet, from the summit of Mount Everest to the deepest oceans [1,2].

Humans consume the equivalent of an entire credit card every week (5 g/week of microplastic) [3]. The health impact of plastic and microplastics on the human body is still unknown [4]. In mice, the analyses of multiple biochemical biomarkers and metabolomic profiles suggested that MP exposure induced the disturbance of energy and lipid metabolism as well as oxidative stress [5].

In a recent study, microplastic particles were revealed in the placentas of unborn babies for the first time, by detecting pigmented microplastics using Raman microscopy [6]. The particles were found in the placentas of healthy women who had normal pregnancies and births. A separate recent study [7] showed that nanoparticles of plastic inhaled by pregnant laboratory rats were detected in the liver, lungs, heart, kidney, and brain of their fetuses.

Polyethylene glycol (PEG) is a hydrophilic polyether compound used in industrial manufacturing in medicine. Although studies indicate the low toxicity of polyethylene glycol on living organisms [8], there are reports of nephrotoxicity [9] damage to the central nervous system and heart, as well as pulmonary and renal failure [10] in PEG-treated animals. Therefore, assessing the impact of polyethylene glycol on the health of humans is essential.

In the present study, we find evidence of PEG exposure in the human placenta, through mass spectrometry analysis. Liquid chromatography–MS/MS technology is a powerful and sensitive technique for the simultaneous characterization and separation of each PEG component [11]. We found the presence of PEG in ten out of twelve examined human placentas, which allows unambiguous identifications based on tandem spectrometry. The MS/MS analysis generates fragments from it to give structural information with high sensitivity and selectivity. The aim of the study is to investigate, in a targeted way, the possible presence in the human placenta of a series of polyethylene glycols (PEGs) and to demonstrate that this xenobiotic particle crosses the placental barrier.

## 2. Results

In our study, an integration of UHPLC coupled with mass spectrometry was used to detect polyethylene glycol (PEG) compounds in 12 human placenta samples. From these 12 placenta samples, three portions (portion 1, portion 2, and membrane portion) were taken from each sample. The three portions of each sample were processed separately for subsequent analysis by UHPLC-MS/MS for a total of 36 individual samples. Among the 12 placentas analyzed, in 10 of them, we identified several fragments of PEGs. For each sample portion, several masses have been annotated. According to Thurman et al. (2014) [12], these masses were related to an ethylene oxidative (EO) from four monomers to 10 (EO-4 to EO-10) PEGs, as shown in Appendix A. In our results, the base peak chromatogram, which only represents the most intense peak of each spectrum of the three portions examined, of sample number “6” shows a signal at the range time from 9 to 10 min, which corresponds to 283.17 *m*/*z* (see Figure 1A).

We found this ion in all of the two portions of the placentas and also in the chorioamniotic membranes. For sample number 6, all three portions had a chromatogram base peak compatible with ion 283.17 *m*/*z*; on the contrary, among the other samples, not all three portions showed this ion—see Table 1, which summarizes the data collected in this study and shows, for each sample, the type of polymer found.

The selected ion (283.17 *m*/*z*) is then activated and fragmented and the fragments are analyzed to generate the MS/MS (MS2) spectrum of the precursor ion. The MS/MS analysis of *m*/*z* 283.1746 at RT 9.15 min highlighted the fragmentation characteristic of PEG-E06. Fragments 177.11–133.09–89.06 *m*/*z*, which differ by 44 units of mass, allow us to unequivocally establish the presence of PEG-EO6 within the analyzed samples (Figure 1B). As also demonstrated by Thurman et al. (2014) [9], the mass was calculated starting from the unit of polyethylene glycol equal to HO-(CH_2_CH_2_O) n-H, which means that the unit chain length must be n = 6, and this length gives the correct formula of C_12_H_26_O_7_ (*m*/*z* 283.17). This pattern matches the chemical structure of PEGs.

Figure 1C shows the total ion current (TIC) and Figure 1D the base peak chromatogram relating to the placenta sample number 4BIS. From these chromatograms, any peaks corresponding to 283.17 *m*/*z* were found. To establish the effectiveness of the plastic-free protocol, analytical blanks were analyzed. The absence of any PEG polymer was found in all of the blanks analyzed, demonstrating the correct execution of the plastic-free protocol and the success of the experiment (Appendix A).

## 3. Discussion

Plastic, microparticles, and their degradation products and the substances they carry could penetrate the human body through the digestive route, the respiratory route, or dermal contact [13,14]. The presence of plastic in its various forms (PEG and microplastics) in the placenta can alter the communication between fetal and maternal cells, which in turn could occur during pregnancy [15,16]. Together with drugs, plastic products and plastic particles can also enter the bloodstream and reach the placenta from the maternal respiratory system [17] or from the gastrointestinal tract (GIT) [18]. The placenta represents the interface between the fetus and the environment [19] and acts as a barrier restricting the access of maternal hormones and xenobiotics to the fetus by enzymatic inactivation or transporting them back into the maternal circulation [20]. The transfer of molecules between maternal and fetal circulation occurs across the endothelial–syncytial membrane of the placenta. Xenobiotic permeation can be influenced by numerous factors: drug properties (degree of ionization, lipophilicity, molecular weight), placental characteristics, and maternal and fetal factors [21]. PEG is a widely used material [22]. Indeed, PEG has been extensively used in drug delivery systems [23] in research [24], industry [25], and diagnostic products [26]. In the present study, for the first time, we identified human PEG exposure, through mass spectrometry analysis, to understand the presence of PEG in placentas. In our results, a series of PEGs have been detected, demonstrating that this xenobiotic particle crosses the placental barrier. In addition, to assess the potential toxicity of PEGs, it is important to distinguish between the monomers and polymers of PEG. Indeed, while PEGs and PEG-Cs are non-toxic [27], the monomer ethylene glycol is considered highly toxic to humans, presenting anti-estrogenic activity [28]. We, therefore, focus on the potential toxicity of low molecular weight PEGs as they consist of fewer monomers (a maximum of six). It is not known whether these are potentially less stable and therefore more prone to degradation into monomers. We found PEG in over 80% of the placentas we tested, but the sample size of this initial study is small and involved patients from a single hospital. Larger studies are needed involving a more diverse patient population to determine how widespread the presence of PEG is in placental tissue. It is also unclear at this time how exposure occurs and whether the health of the fetus is adversely affected by its presence. In utero, fetal health can be put at risk when MPs disrupt the lines of maternal–fetal communication and the transport of important nutrients. Preeclampsia and fetal growth restriction may even develop [29,30].

In this study, high-resolution mass spectrometry was used to identify mono- and polymers of PEGs, therefore it is necessary to continue with a metabolomics analysis to investigate the influence on the metabolism by the low molecular weight polymers detected, comparing the samples of the placenta in which the presence of high and low molecular weight PEG was detected with samples in which no plastic particles were detected.

## 4. Materials and Methods

### 4.1. Enrolment of Patients and Placentas Collection

Twelve women who gave birth at the “San Giovanni Calibita” Fatebenefratelli Hospital (Rome) in the period from December 2019 to December 2021 were enrolled. All participants provided informed written consent to participate in the research project, and the study was approved by the Ethical Committee Lazio 1 (Prot. N. 352/CE Lazio1; 31 March 2020).

The placentas were collected from both caesarean section and spontaneous deliveries via the vagina.

Patients were excluded from the study according to the following criteria:-Taking special diets prescribed for medical reasons within 4 weeks before childbirth.-Diarrhea or constipation in the 2 weeks before giving birth.-Taking antibiotics in the 2 weeks prior to childbirth.-Taking medications that affect intestinal resorption (e.g., activated charcoal or cholestyramine) in the 2 weeks before childbirth.-Diagnosis of gastrointestinal disease (e.g., ulcerative colitis or Crohn’s disease, excluding appendectomy), cancer, organ transplant, HIV, or any other serious illness that led to medical treatment.-Invasive or abrasive dental treatments in the 2 weeks prior to childbirth.-Current or recent participation (within 4 weeks before delivery) in a clinical trial.-Alcohol abuse (defined as Alcohol Use Disorder Identification Test Score >10) [31].

The sampling of the placentas was performed according to a specific protocol to avoid contamination with plastic or synthetic fibers.

Routine procedures were used for plastic-free removal of the placenta during cesarean section and during vaginal delivery. The operating room for the cesarean section was set up according to traditional methods, taking care to affix a cotton cloth to cover the lower part of the bag for the collection of blood losses. The operators wore cotton gloves to perform the surgery. After the fetus was expelled, the umbilical cord was blocked with metal pliers, and once cut, the cord was placed on the cloth described above, taking care not to put it in contact with plastic parts. The operators proceeded to give birth by squeezing, taking care not to insert the hand inside the uterine cavity.

In the case of vaginal birth, the assistance techniques were the usual, with the difference that the midwife wore cotton gloves during the final stages of the expulsive period. The camp was set up by placing a cotton cloth under the maternal buttocks; a graduated bag was not used, which was positioned only after the placenta. After the fetus was expelled, the umbilical cord was blocked with metal pliers, and once cut, the cord was placed on the cloth described above, taking care not to put it in contact with plastic parts. After delivery, the placentas were placed in a special metal container and transported to the pathological anatomy laboratory. The pathologist wore cotton gloves to perform, with a special metal scalpel, the sections of the placenta and the amnio chorial membranes. The samples obtained were placed in special plastic-free glass containers and stored at a temperature of −20 °C.

### 4.2. Plastic-Free Extraction Protocol

From each placenta, three portions with a mean weight of 1g were randomly collected from the placenta and the chorioamniotic membranes. Each portion was extracted following a plastic-free protocol. In detail, the placentas were placed in a metal mixer to obtain a thoroughly mixed product and transferred into a glass tube. Then, 5 mL of ice-cold ultra-pure water (18 MQ) was added using glass Pasteur pipettes to lyse the cells. The tubes were plunged into dry ice or a circulating bath at −25 °C for 0.5 min and then into a water bath at 37 °C for 0.5 min and then sonicated for another 2min. To each tube, 7 mL of −20 °C methanol and then 7mL of −20 °C chloroform was added. The tubes were mixed every 5 min for 30 min before being transferred to −20 °C for 2–8 h. The next day, 0.5 mL from each phase was withdrawn and placed in 1 mL glass tubes and the collected supernatants were dried at 60 °C overnight. Finally, the dried samples were re-suspended in 0.2 mL of water and transferred to glass autosampler vials for LC-MS analysis.

The analytical blank was prepared using the same plastic-free workflow used for the samples. That is, using glass tubes, glass Pasteur pipettes, and glass autosampler vials for LC-MS analysis. The analytical blank contained ultra-pure water (18 MQ) with the addition of 0.1% formic acid (the same formula used for solvent A).

### 4.3. UHPLC-MSMS Analysis

The supernatants were injected into an UHPLC system (Ultimate 3000, Thermo Thermo Fisher Scientific, 168 Third Avenue Waltham, MA, USA 02451) and run in positive ion mode. A Reprosil C18 column (2.0 mm × 150 mm, 2.5 μm—Dr. Maisch, Germany) was used for separation. Chromatographic separations were achieved at a column temperature of 30 °C and a flow rate of 0.2 mL/min. The optimized chromatographic method starts with the initial mobile phase composition of a 100% linear gradient of solvent A (ddH_2_O, 0.1% formic acid), followed by a linear gradient to 100% B (acetonitrile, 0.1% formic acid) after 20 min returning to 100% A in 3 min. A combination of ultra-high-performance liquid chromatography (UHPLC) coupled with Q-Exactive (Thermo) mass spectrometry (MS/MS) was used to detect the distribution of polyethylene glycols. The instrument was used in positive ionization mode with full scan (FS) and a subsequent data-dependent acquisition (DDA) mode. The settings for the full scan were as follows: resolution, 70,000; automatic gain control (AGC) target, 3e6; maximum injection time (IT), 100 ms; and scan range, *m*/*z* 70–1050. The remaining settings for DDA mode were as follows: resolution, 17,500; AGC target, 1e5; maximum IT, 50 ms; isolation window, *m*/*z* 1.0, collision energy (NCE), 30. The stability of mass accuracy was checked daily and if the values went above 2 ppm error, then the instrument was re-calibrated. Calibration was performed before each analysis against positive or negative ion mode calibration mixes (Pierce, Thermo Fisher, Rockford, IL, USA) to ensure the error of the intact mass within the sub-ppm range. Data acquisition and data processing were achieved using Thermo Scientific™ Xcalibur™ software. A standard was not available for the different types of PEGs monomers (EO-4 to EO-10), therefore, we only conducted qualitative rather than quantitative analyses.

## 5. Conclusions

In conclusion, this study aimed to evaluate the level of human placenta exposure to PEG. Due to the placenta’s crucial role in supporting fetal development and acting as an interface between the fetus and the external environment, the presence of exogenous and potentially harmful PEG compounds is of great concern. The presence of PEG in ten of twelve of the placentas under examination and the fact that, when present, PEG is capable of spreading to any part of the examined placentas (portion 1, portion 2 from the placenta, and the membrane portion from the chorioamniotic portion) confirms that the PEG, once inside the human body, reaches the tissues of the placenta at all levels and could alter the interconnection between mother and fetus. However, our research is very preliminary, so there are several limitations. One limitation is the number of samples, so more samples will be needed to follow up on the measured metabolite. Moreover, although we identified PEG in ten of the twelve placentas under examination, other future investigations, with other complementary technologies, are necessary for identifying the presence of other MPs and for the estimation of the total amount of MPs in the human placenta. Finally, further studies need to be conducted to understand whether the presence of PEG in the human placenta can trigger immune responses or lead to the release of toxic contaminants, which are harmful to pregnancy.

The data obtained in this study will be used, in a later study, for an untargeted metabolomic analysis to highlight any changes in metabolism in the studied placenta samples. In this way, it will be possible to understand the effects of PEG in the human placenta, by identifying the metabolic pathways that may be compromised, by comparing the metabolic profiling of the samples in which plastic polymers were detected and those in which they were not detected.

## Figures and Tables

**Figure 1 ijms-23-12743-f001:**
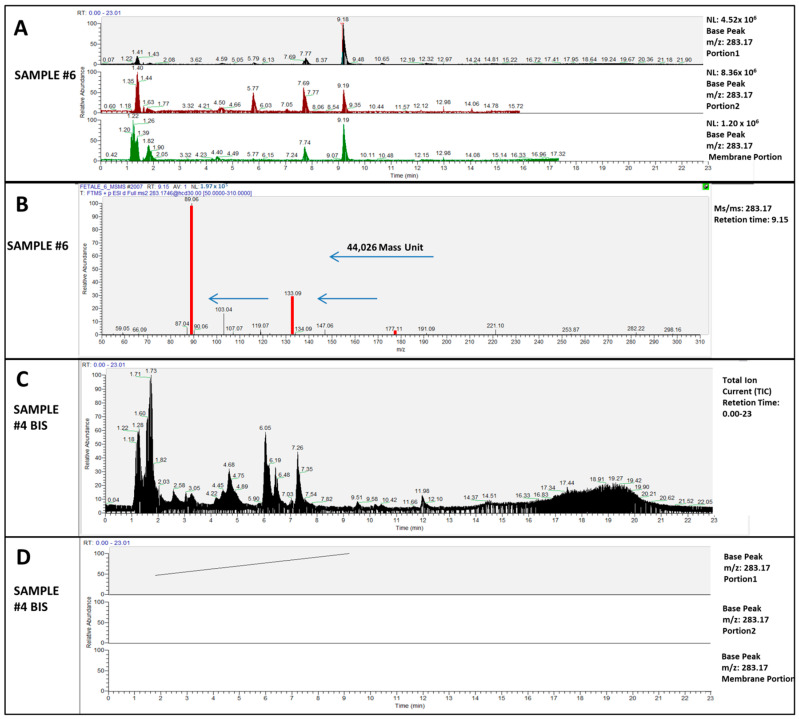
(**A**): Base peak chromatogram at RT 9.15 min of 283.17 *m*/*z* of all portions collected from sample #6. (**B**): A MS/MS spectrum showing the differences in mass of the polyethylene glycol series (44.0262 mass units). (**C**): Total ion current (TIC), of Placenta #4bis considered control samples. (**D**): Base peak chromatogram of 283.17 *m*/*z* of three portions collected from sample Placenta #4bis. The base peak chromatogram does not report any peaks corresponding to 283.17 *m*/*z*.

**Table 1 ijms-23-12743-t001:** Identification of different masses of ethylene oxidative with 4 monomers to 10 monomers (EO-4 to EO-10) of PEGs in portion 1 and portion 2 from the placenta and the membrane portion. The symbol “*” represents the presence of the PEG fragment in the sample.

*Putative* *Formula*	*Putative Identification*	*Calculated* *Exact Mass*	*Portions*		*1bis*	*2bis*	*3*	*3bis*	*4*	*4bis*	*5*	*6*	*7*	*8*	*9*	*10*
**C_12_H_26_O_7_**	**PEG-EO6**	**283.1752**	** *Placenta* **	** *Portion 1* **	*****	*****	*****	*****	*****	**-**	*****	*****	*****	*****	*****	**-**
				** *Portion 2* **	*****	*****	*****	*****	*****	**-**	**-**	*****	*****	*****	*****	**-**
			** *Chorioamniotic* **	** *Membrane Portion* **	*****	*****	*****	*****	*****	**-**	*****	*****	*****	*****	**-**	**-**
**C_14_H_30_O_8_**	**PEG-EO7**	**327.2013**	** *Placenta* **	** *Portion 1* **	*****	*****	**-**	*****	**-**	**-**	*****	*****	*****	**-**	**-**	**-**
				** *Portion 2* **	*****	*****	*****	*****	*****	**-**	*****	*****	*****	*****	*****	**-**
			** *Chorioamniotic* **	** *Membrane Portion* **	**-**	*****	**-**	*****	*****	**-**	*****	**-**	**-**	**-**	**-**	**-**
**C_16_H_34_O_9_**	**PEG-EO8**	**371.2276**	** *Placenta* **	** *Portion 1* **	*****	*****	*****	*****	**-**	**-**	*****	*****	*****	*****	**-**	**-**
				** *Portion 2* **	*****	*****	*****	**-**	*****	**-**	*****	*****	**-**	*****	*****	**-**
			** *Chorioamniotic* **	** *Membrane Portion* **	**-**	*****	**-**	*****	*****	**-**	*****	*****	*****	**-**	**-**	**-**
**C_18_H_38_O_10_**	**PEG-EO9**	**415.2538**	** *Placenta* **	** *Portion 1* **	**-**	*****	**-**	**-**	**-**	**-**	*****	*****	**-**	*****	**-**	**-**
				** *Portion 2* **	**-**	*****	**-**	*****	*****	**-**	*****	**-**	*****	*****	*****	**-**
			** *Chorioamniotic* **	** *Membrane Portion* **	**-**	**-**	**-**	**-**	*****	**-**	*****	**-**	**-**	**-**	*****	**-**
**C_20_H_42_O_11_**	**PEG-EO10**	**459.2800**	** *Placenta* **	** *Portion 1* **	**-**	*****	**-**	**-**	**-**	**-**	*****	**-**	*****	**-**	**-**	**-**
				** *Portion 2* **	**-**	**-**	**-**	*****	**-**	**-**	*****	**-**	*****	*****	*****	**-**
			** *Chorioamniotic* **	** *Membrane Portion* **	**-**	**-**	**-**	**-**	**-**	**-**	*****	*****	**-**	**-**	*****	**-**

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
