# Peer review of "Plastic and Placenta: Identification of Polyethylene Glycol (PEG) Compounds in the Human Placenta by HPLC-MS/MS System"

_ijms, 2022, doi:10.3390/ijms232112743_

Round 1
Reviewer 1 Report
Title: Plastic and Placenta: Identification of polyethylene glycol (PEG) compounds in the human placenta by HPLC-MS/MS system
Manuscript ID: ijms-1914723
Type: Brief Report
Comments and Suggestions for Authors
This study used Ultra-High-Performance Liquid Chromatography (UHPLC) coupled by Mass Spectrometry (MS) to detect suites of polyethylene glycol (PEG) compounds in human placenta samples. The aim of the study was to investigate are PEGs present in human placenta and do the cross the placental barrier. PEGs were found, for the first time, in 10 out of 12 placentas and they were shown to cross the placental barrier.
Abstract
Abstract is concise and well written. However, it remains unclear what for PEG in placentas was studied. (What are the health risks of PEG for the fetus/newborn? Why was PEG chosen as a target of investigation?)
Introduction
Introduction gives background for the study. Aim of the study is clearly stated.
Results
Results are clearly presented.
Discussion
Is there any evidence that PEG and microplastics affect the function of the placenta? Is there a reference for “The presence of plastic in its various forms (PEG and microplastics) in the placenta can alter the communication between fetal and maternal cells which in turn could occur during pregnancy.” ?
Authors state that PEG is a widely used material but could they elaborate a little: what are the most common applications? How do PEGs end up in maternal circulation?
“Even though PEG is generally considered biologically inert and safe in animals and humans, the slow clearance of large PEGs raises concerns about potential adverse effects resulting from PEG accumulation in tissues following chronic administration, particularly in the central nervous system [19].”
Is there any evidence that PEGs could be harmful for central nervous system? What kind of effects PEGs could have? Do they cross blood-brain barrier? Is CNS toxicity just speculation? If so, it should be removed.
It is better reasoned why authors focused on low molecular weight PEGs.
Authors do discuss the weaknesses of the paper (low number of cases, single hospital).
It is claimed that MPs could be a risk for fetal well being but is there any evidence of this? Is there any estimation on the total amount of MPs in human placenta? (Placental insufficiency starts affecting fetal well-being when approximately 50% of placental function has been lost and therefore it is a little hard to believe MPs could have such a big impact.)
Conclusions
Maybe future studies will answer the questions raised by this study.
Materials and methods
12 women were enrolled during one year to the study. Can the authors elaborate why such a long time was needed to recruit quite few participants?
Can authors speculate why PEGs were not found in all the placentas? Is it a matter of technique used or could there be differences in mothers diet?
Methods are clearly explained.
Author Response
Manuscript ID: ijms-1914723
Authors: Antonio Ragusa, Veronica Lelli, Giuseppina Fanelli, Alessandro Svelato, Sara D’Avino, Federica Gevi, Criselda Santacroce, Piera Catalano, Mauro Ciro Antonio Rongioletti, Caterina De Luca, Alessandra Gulotta, Sara Rinalducci, Anna Maria Timperio.
Title: Plastic and Placenta: Identification of polyethylene glycol (PEG) compounds in the human placenta by HPLC-MS/MS system
Review 1
This study used Ultra-High-Performance Liquid Chromatography (UHPLC) coupled by Mass Spectrometry (MS) to detect suites of polyethylene glycol (PEG) compounds in human placenta samples. The aim of the study was to investigate are PEGs present in human placenta and do the cross the placental barrier. PEGs were found, for the first time, in 10 out of 12 placentas and they were shown to cross the placental barrier.
- Abstract is concise and well written. However, it remains unclear what for PEG in placentas was studied. (What are the health risks of PEG for the fetus/newborn? Why was PEG chosen as a target of investigation?)
Author replay: We are thankful to review for the interesting considerations. We would like to point out that the main objective of our work was to investigate through, LC-MS analysis, the presence, in human placenta, of total microplastic particles. In this first study we found the presence of PEGs only. In the new version we added new references on health risks of PEG for fetus/newborn (see below).
- Is there any evidence that PEG and microplastics affect the function of the placenta? Is there a reference for “The presence of plastic in its various forms (PEG and microplastics) in the placenta can alter the communication between fetal and maternal cells which in turn could occur during pregnancy.” ?Authors state that PEG is a widely used material but could they elaborate a little: what are the most common applications? How do PEGs end up in maternal circulation?
Author replay: The Authors would like to thank the reviewer for his/her constructive suggestions. So we add some references that confirm that maternal MPs exposure induced the metabolic disorders in their offspring (lines 108 to 110) the wide range of applications of poly(ethylene)glycol (PEG) (lines 120-121)and How do PEGs end up in maternal circulation (lines110 to112).
1) Luo T, Zhang Y, Wang C, et al. Maternal exposure to different sizes of polystyrene microplastics during gestation causes metabolic disorders in their offspring. Environ Pollut. 2019;255(Pt 1):113122. doi:10.1016/j.envpol.2019.113122; 2) Zaheer J, Kim H, Ko IO, et al. Pre/post-natal exposure to microplastic as a potential risk factor for autism spectrum disorder. Environ Int. 2022;161:107121. doi:10.1016/j.envint.2022.107121; 3) Schlesinger PH., Krogstad DJ., Herwaldt BL., 1988. Antimalarial agents: mechanisms of action. Antimicrob Agents Chemother. 32(6):793-8. doi: 10.1128/AAC.32.6.793. PMID: 3046479; PMCID: PMC172284; 4) Arumugasaamy N., Navarro J., Kent Leach J., Kim PCW., Fisher JP., 2019. In Vitro Models for Studying Transport Across Epithelial Tissue Barriers. Ann Biomed Eng. ;47(1):1-21. doi: 10.1007/s10439-018-02124-w. Epub 2018 Sep 14. PMID: 30218224; 5) D'souza AA, Shegokar R. Polyethylene glycol (PEG): a versatile polymer for pharmaceutical applications. Expert Opin Drug Deliv. 2016;13(9):1257-1275. doi:10.1080/17425247.2016.1182485; 6) Dai Q, Walkey C, Chan WCW (2014) Polyethylene glycol backfilling mitigates the negative impact of the protein corona on nanoparticle cell targeting. Angew Chem Int Ed Engl 53:5093–5096; 7) Kianpour E, Azizian S (2014) Polyethylene glycol as a green solvent for effective extractive desulfurization of liquid fuel at ambient conditions. Fuel 137:36–40; 8) Minordi LM, Vecchioli A, Mirk P, Bonomo L (2011) CT enterography with polyethylene glycol solution vs CT enteroclysis in small bowel disease. Br J Radiol 84:112–119
- “Even though PEG is generally considered biologically inert and safe in animals and humans, the slow clearance of large PEGs raises concerns about potential adverse effects resulting from PEG accumulation in tissues following chronic administration, particularly in the central nervous system [19].” Is there any evidence that PEGs could be harmful for central nervous system? What kind of effects PEGs could have? Do they cross blood-brain barrier? Is CNS toxicity just speculation? If so, it should be removed.
Author replay: We thank you and agree with the reviewer for the comment that we consider appropriate. Although the slow clearance of large PEGs raises concerns about potential adverse effects, particularly in the central nervous system, there is limited information on the real impact of PEGs on the possibility to cross blood-brain barrier, so a whole and more PEG toxicity research is needed to truly understand this compound’s effects on the human body in particular for newborn or fetus, taking advice from the reviews the sentences are now omitted.
- It is better reasoned why authors focused on low molecular weight PEGs.
Author replay: Each compound of PEGs increases by the chemical attachment of ethylene glycol unit, which means an accurate adding molecular mass of 44.0262. Thus, in our study we obtained the correct formula and structure for one of the glycols, consequently the others can be calculated adding or subtracting the 44.0262 mass unit. For the purpose of the study, it was important that at least one of the compounds be identified with its ms/ms. So we focused our attention on the first ms/ms identified compound (PEG-EO6) with a characteristic fragmentation pattern, with a series peak separated by 44.02 mass units (ethylene oxide group) [CH2CH-O-] that unequivocally was used as a method to indicate the presence of PEG.
- Authors do discuss the weaknesses of the paper (low number of cases, single hospital). It is claimed that MPs could be a risk for fetal well being but is there any evidence of this? Is there any estimation on the total amount of MPs in human placenta? (Placental insufficiency starts affecting fetal well-being when approximately 50% of placental function has been lost and therefore it is a little hard to believe MPs could have such a big impact.)
Author replay: The suggestion is stimulating, but the work is focused on detecting suites of PEG compounds in human placenta samples and the estimation on the total amount of MPs in human placenta requires remarkably different experimental approaches and methods. However, the authors take into consideration the suggestion of the referee and dedicate a section of conclusion to the drawbacks and limitations of the study (lines 240 to 246)
- 12 women were enrolled during one year to the study. Can the authors elaborate why such a long time was needed to recruit quite few participants?
Author replay: For this study, we recruited only healthy women that have a vaginal delivery at term of pregnancy. They were selected according to specific exclusion criteria so the number of samples was significantly reduced during the recruitment. Moreover, to prevent plastic contamination, a plastic-free protocol was adopted during the entire experiment. Obstetricians and midwives used cotton gloves to assist women in labor. In the delivery room, only cotton towels were used to cover patients' beds; graduate bags to estimate postpartum blood loss were not used during delivery, but they were brought in the delivery room only after birth, when umbilical cord was already clamped and cut with metal clippers, avoiding contact with plastic material. Pathologists wore cotton gloves and used metal scalpels. All these aspects have not always made it possible to use the placentas of donors.
Can authors speculate why PEGs were not found in all the placentas? Is it a matter of technique used or could there be differences in mothers' diet?
Author replay: The absorption of PEG can depend on different physiological conditions and genetic characteristics; this could explain, together with the different eating habits and lifestyles of the patients, the absence of PEG particles in 2 of the 12 analyzed placentas. Moreover, it is known that there is great variability in the expression and function of placental drug transporters, both within human populations (inter-individual variability) and during gestation (intra-individual variability) (Cihalova D., Ceckova M., Kucera R., Klimes J., Staud F., 2015. Dinaciclib, a cyclin-dependent kinase inhibitor, is a substrate of human ABCB1 and ABCG2 and an inhibitor of human ABCC1 in vitro. Biochem Pharmacol. 1;98(3):465-72. doi: 10.1016/j.bcp.2015.08.099.) Assuming that this variability also exists in relation to the mechanism of internalization of PEG, these can potentially explain the presence of these exogenic materials in the placenta during pregnancy. However, as a pilot study, we would like to point out the presence in human placenta of polyethylene glycols (PEG) demonstrating, for the first time, that this xenobiotic particle crosses the placental barrier.
Author Response
Manuscript ID: ijms-1914723
Authors: Antonio Ragusa, Veronica Lelli, Giuseppina Fanelli, Alessandro Svelato, Sara D’Avino, Federica Gevi, Criselda Santacroce, Piera Catalano, Mauro Ciro Antonio Rongioletti, Caterina De Luca, Alessandra Gulotta, Sara Rinalducci, Anna Maria Timperio.
Title: Plastic and Placenta: Identification of polyethylene glycol (PEG) compounds in the human placenta by HPLC-MS/MS system
This is a brief report on the identification of polyethylene glycol (PEG) compounds in the human placenta by the use of Ultra-High-Performance Liquid Chromatography coupled by Mass Spectrometry as a new technique. The sample size is appropriate to draw a preliminary conclusion. Introduction The introduction is clear. Materials and Methods should be the next paragraph in the script.
The authors did not specify in the exclusion criteria if they take into account pregnancies that undergo invasive procedure such as chorionic villous sampling or amniocentesis?
Results well described. Discussion is clear.
This is an excellent work. The manuscript is very well written and is easy to read, the methodology is well described, the results are clearly exposed, and the findings well discussed, including limitations of the study. I have no criticism to this manuscript, and in my opinion the study is suitable for publication.
Author replay: The authors would like to thank this Reviewer for his/her kind comments. We are very glad that the Reviewer positively evaluated our study planning and methodology used for reaching our results. Regarding the exclusion criteria, we recruited only healthy women that have a vaginal delivery at term of pregnancy and not subjected to invasive procedures such as chorionic villus sampling or amniocentesis. They other exclusion criteria were: diagnosis of gastrointestinal disease, such as ulcerative colitis, or Crohn’s disease, cancer, organ transplantation, HIV (Human Immunodeficiency Virus), or other severe pathologies; alcohol abuse (defined as a >10 score in the Alcohol Use Disorders Identification Test); cigarette smoking; peculiar diets prescribed for particular medical conditions (four weeks before delivery); diarrhea or constipation (two weeks before delivery); antibiotics intake (two weeks before delivery); assumption of drugs affecting intestinal reabsorption, such as activated charcoal, or cholestyramine (two weeks before delivery); invasive or abrasive dental treatments (two weeks before delivery); participation to a clinical study (four weeks before delivery). (Ragusa A. Svelato A. Santacroce C. Catalano P. Notarstefano V. Carnevali O. Papa F. Rongioletti MCA. Baiocco F. Draghi S. D'Amore E. Rinaldo D. Matta M. Giorgini E. Plasticenta: First evidence of microplastics in human placenta. Environment international 2021, 146: 106274. doi: 10.1016/j.envint.2020.106274).
Reviewer 3 Report
The manuscript by Ragusa et al presents an analysis of the presence of PEG-derived compounds in samples from placentas obtained from normal pregnancies. By using a combination of UHPLC and MS the authors show that PEG-derived compounds are present in the tissue samples analyzed.
The manuscript is well written, some issues require clarification.
The MS method used to identify PEG-derived compounds lacks controls, no internal controls were used, and the authors rely only in mass difference from MS to identify PEG-derivatives containing between 6-10 etoxylated units. Usually heavy internal controls give a specific answer. Although the authors claim this to be a qualitative and not quantitative study, the way the results are presented seem a weak support for identification of PEG-derivatives compounds. A NMR analysis woud provide unequivocal identification.
How sample# 4BIS is considered a control? A total ion current (TIC) chromatogram is shown for sample #4BIS and is used to claim the absence of peak at RT 9.15 min/ 283.17 m/z as further confirmation of the identification of PEG-derivatives. The difference bewteen sample #BIS and others is not explained.
Author Response
Manuscript ID: ijms-1914723
Authors: Antonio Ragusa, Veronica Lelli, Giuseppina Fanelli, Alessandro Svelato, Sara D’Avino, Federica Gevi, Criselda Santacroce, Piera Catalano, Mauro Ciro Antonio Rongioletti, Caterina De Luca, Alessandra Gulotta, Sara Rinalducci, Anna Maria Timperio.
Title: Plastic and Placenta: Identification of polyethylene glycol (PEG) compounds in the human placenta by HPLC-MS/MS system
The manuscript by Ragusa et al presents an analysis of the presence of PEG-derived compounds in samples from placentas obtained from normal pregnancies. By using a combination of UHPLC and MS the authors show that PEG-derived compounds are present in the tissue samples analyzed.
The manuscript is well written, some issues require clarification.
- The MS method used to identify PEG-derived compounds lacks controls, no internal controls were used, and the authors rely only in mass difference from MS to identify PEG-derivatives containing between 6-10 etoxylated units. Usually heavy internal controls give a specific answer. Although the authors claim this to be a qualitative and not quantitative study, the way the results are presented seem a weak support for identification of PEG-derivatives compounds. A NMR analysis woud provide unequivocal identification.
Author replay: The authors know that the reviewer's suggestion is very useful. Our intent was to investigate, in a targeted way, the presence in the human placenta of PEG since previous studies carried out by our group have shown (through Raman microspectroscopy) the accumulation of microplastic particles on this organ. Mass spectrometry (MS) and Nuclear Magnetic Resonance (NMR) are the two most widely used techniques for structural identification of compounds:both NMR and MS can be used to detect and identify metabolites and measure their concentrations accurately. Obviously the combination of more techniques perform a comprehensive analysis of these xenobiotic metabolites, so future research using different experimental approaches will be performed.
- How sample# 4BIS is considered a control? A total ion current (TIC) chromatogram is shown for sample #4BIS and is used to claim the absence of peak at RT 9.15 min/ 283.17 m/z as further confirmation of the identification of PEG-derivatives. The difference bewteen sample #BIS and others is not explained.
Author replay: The figure 1 shows two samples called sample #6 (fig. 1A and 1B) and #4bis (Fig. 1C and 1D). For each of them we show both total ion current (TIC) and base peak chromatogram (BPC). We considered sample #4bis as control because the base peak chromatography do not show any peak corresponding to 283.17 m/z in none of the portions taken in consideration so we conclude that in this sample there is no PEG and for these reasons sample #4BIS is considered a control. Concerning sample nomenclature there is not a specific reason and it just depends on the samples recruitment made by the “San Giovanni Calibita” Fatebenefratelli Hospital (Rome).